# Online Monitoring of Sourdough Fermentation Using a Gas Sensor Array with Multivariate Data Analysis

**DOI:** 10.3390/s23187681

**Published:** 2023-09-06

**Authors:** Marvin Anker, Abdolrahim Yousefi-Darani, Viktoria Zettel, Olivier Paquet-Durand, Bernd Hitzmann, Christian Krupitzer

**Affiliations:** 1Department of Food Informatics and Computational Science Hub, University of Hohenheim, 70599 Stuttgart, Germany; marvin.anker@uni-hohenheim.de; 2Department of Process Analytics and Cereal Science, University of Hohenheim, 70599 Stuttgart, Germany; rahim_yousefi2002@yahoo.com (A.Y.-D.); o.paquet-durand@uni-hohenheim.de (O.P.-D.); bernd.hitzmann@uni-hohenheim.de (B.H.)

**Keywords:** gas sensor, machine learning, process analytics, process modeling, food monitoring, sourdough

## Abstract

Sourdough can improve bakery products’ shelf life, sensory properties, and nutrient composition. To ensure high-quality sourdough, the fermentation has to be monitored. The characteristic process variables for sourdough fermentation are pH and the degree of acidity measured as total titratable acidity (TTA). The time- and cost-intensive offline measurement of process variables can be improved by utilizing online gas measurements in prediction models. Therefore, a gas sensor array (GSA) system was used to monitor the fermentation process of sourdough online by correlation of exhaust gas data with offline measurement values of the process variables. Three methods were tested to utilize the extracted features from GSA to create the models. The most robust prediction models were achieved using a PCA (Principal Component Analysis) on all features and combined two fermentations. The calibrations with the extracted features had a percentage root mean square error (RMSE) from 1.4% to 12% for the pH and from 2.7% to 9.3% for the TTA. The coefficient of determination ($R^{2}$) for these calibrations was 0.94 to 0.998 for the pH and 0.947 to 0.994 for the TTA. The obtained results indicate that the online measurement of exhaust gas from sourdough fermentations with gas sensor arrays can be a cheap and efficient application to predict pH and TTA.

## 1. Introduction

Sourdough is one of the oldest examples of natural starters, mainly used for making fermented baked goods as an alternative to baker’s yeast and chemical leavening. Sourdough fermentation is a unique tool for improving the rheology, sensory properties, shelf life, and nutrient composition of gluten-free formulation [1]. The quality and properties of sourdough depend on many technological and ecological influences. The most critical factors for sourdough production are contributed by the contents and enzymatic activity of the cereal, the controllable process parameters, and the microflora of lactic acid bacteria, yeasts, and other microorganisms [2]. Sourdough fermentation is characterized by the process variables’ pH, which is essential for inhibiting enzyme activity, and the degree of acidity measured as total titratable acidity (TTA), which accounts for the evaluation of the sensory properties.

The ability of gas sensors to measure specific gases like O_2_, CO_2_, and hydrogen sulfides which are linked to spoilage is a critical topic with respect to the attributes of quality, freshness, and safety conditions [3]. Fermentation monitoring uses gas sensor arrays that combine specific gas sensor signals with pattern recognition [4]. The sensor is set between qualitative (i.e., improve sensory attributes) or quantitative (i.e., monitoring) measurement [5] depending on the research goal. Research on fermentation monitoring with electronic nose techniques splits into submerged and solid-state fermentation. The submerged fermentation is relevant to our research, which includes all fermentations in the presence of excess water. Here, the electronic nose mainly uses fingerprints obtained from odor [1]. Pinheiro et al. [6] investigated aroma production with an electronic nose for monitoring the fermentation process by using the unspecific sensor signal corresponding to ethanol concentration. Zhang et al. [7] investigated fermentation monitoring by alcoholic quantification with the use of an electronic nose as well as near-infrared spectroscopy. Several studies use gas sensor arrays to monitor fermentation, like Genzardi et al. [8] and Oikonomou et al. [9]. Monitoring these process variables offline is a cost- and time-consuming procedure. Grote et al. [10] used fluorescence spectroscopy to monitor the sourdough fermentation process online. In the study by Bolarinwa et al. [11], the influence of processing conditions on the levels of pH and TTA was determined in rice sourdough. They developed a prediction model that could predict the response of pH and TTA. A cheap and effective alternative can be monitored with a gas sensor array (GSA) system. With increased performance and availability, the properties of GSA, like a fast assessment of headspace and the ability for quantitative representation of gas mixtures, bode well for monitoring tasks. This proved to be especially useful for microbial fermentation monitoring with the analysis of exhaust gas [12]. The challenge of measuring pH during sourdough fermentation is that it consists of liquid and solid parts that can influence the electrode. A reliable result can only be achieved by diluting a sample. Therefore, predicting critical measurement parameters like pH and acidity with a gas sensor array (GSA) system can be a cheap and effective alternative. Electronic noses have been widely established for the determination of the chemical composition in fermented foods and beverage application [13,14]. Here, they are used for sensory evaluation, for example, by identifying essential aroma compounds that are responsible for staling of bread [15] or qualitative analysis to detect product adulteration like substituting corn and rice syrup in honey [16]. However, due to their unspecificity, they need to improve their ability for quantitative analysis. Using soft sensor models improves the effectiveness and possible application areas for GSA systems. Many approaches to fermentation monitoring with soft sensors, including data processing techniques, such as multiple least square support vector machine, neural network, deep learning, fuzzy logic, and probabilistic latent variable models, have been collected by Zhu et al. [17]. Viejo et al. [18] showed that with machine learning, it was possible to build highly accurate and precise models to determine the type of wheat and the volatile components of sourdough bread. Mei et al. [19] used a practical soft sensor modeling approach that combines PCA and Gaussian process regression to predict the biomass concentration in a fermentation better than a neural network and support vector machine model.

The previous studies correlate the gas sensor output to certain process variables or aroma characteristics. These studies visualize the kinetics of the associated target. This paper presents a GSA system for evaluating the online prediction capability of the process variables pH and TTA. Our approach aims to create prediction models that allow intrapolation and, therefore, enable automating monitoring tasks for sourdough fermentation. This should be achieved by feeding the models large amounts of data on different temperature and flour type instances. Therefore, the prediction models should be ubiquitously usable for sourdough fermentation. We carried out experiments with three different temperatures and various types of flour to investigate the performance of our analysis approach. Consequently, we provide the following contributions:A multivariate data analysis approach for the sourdough fermentation process.A correlation of features from the online GSA measurement values with offline measurements of the process variables.Creation of prediction models with a parametric regression approach.

The remainder of this paper is structured as follows. In Section 2, we explain how sourdough fermentation works, describe the gas sensor array system, and illustrate the procedure for the experiments. Section 3 shows the obtained results. Finally, Section 4 discusses the achieved results, and Section 5 puts them into context and shows further possibilities.

## 2. Methods and Materials

First, Section 2.1 explains the sourdough fermentation process. After, we describe the working principle and application of the gas sensor array system in Section 2.2. Last, Section 2.3 illustrates the procedure for the experiments.

### 2.1. Sourdough Fermentation

Sourdough is a fermented dough with microorganisms, primarily lactic acid bacteria and active or reactivated yeasts. The acidification of the dough is only obtained by fermentation. During the fermentation, sugars are cleaved into carbon dioxide, which is incorporated in the dough to increase its volume and small amounts of alcohol and aroma components, which include lactic and acetic acid. Fresh sourdough is initiated by mixing flour and water and leavening it at a warm temperature. After 12–24 h, spontaneous fermentation leads to an acidic and alcoholic odor of the mixture. The TTA can provide a simple estimate of the total acid content but cannot differentiate the acids within the food sample. Whereas the TTA is a better predictor of acid’s impact on flavor than pH, the pH can better describe how well microorganisms can grow in a food matrix due to the dependence on hydronium ion concentration [1]. Hence, both measurements might be relevant to control the fermentation process.

Besides the dough acidity, the dough yield (DY) is an essential parameter for characterizing sourdoughs. It describes the dough consistency as the ratio between water and flour in the dough. It is calculated with the following formula [20] (see Equation (Equation 1)):(1)$$DY=\frac{(m(flour)[g]+m(water)[g])\times 100}{m(flour)[g]}$$

The same *DY* in sourdoughs does not mean that they have the same consistency, because different flours have different abilities to absorb water. Generally, sourdoughs with a *DY* of 150–160 have a firm consistency, and doughs with a *DY* of 200 begin to show a liquid consistency [20]. Also, the acidification rate increases with higher *DY* due to the enhanced diffusion of components in the dough with increased fluidity. Faster acidification of the dough means that the fermentation times are reduced as well [21]. In this work, we investigated different doughs based on different flour types and different *DY* characteristics.

### 2.2. Gas Sensor Array System

We measured online exhaust gas with a self-assembled measurement system [22]. Figure 1 shows the setup for the GSA system. The electricity for the GSA setup is provided by a multifunction AC/DC-voltage source (1). The exhaust gas reaches the GSA setup from the connection tube (2) of the fermenter. The exhaust gas is led to the gas chamber (3), which is also connected to a gas flow meter (4) that receives oxygen from a gas flask that is connected by a tube (5) to the setup. The signal measured in the gas chamber is transferred to the Arduino mega 2560 (6) that forwards the signal to the Matlab script on the connected laptop.

Figure 2 provides a schematic view of the GSA system and how it is integrated into the fermentation setup. The measurement system contains two main parts: the headspace sampling system and the measurement chamber. The headspace sampling procedure consisted of an automated sequence of internal operations. First, the headspace samples of the fermenter are pumped past the measurement chamber for 10 s at a flow rate of 600 mL/min with a diaphragm pump (Schwarzer Precision) every five minutes. The measurement chamber has a volume of 250 mL and contains a gas sensor array equipped with commercially available metal oxide semiconductor (MOS) gas sensors (TGS 822, TGS 813, and MQ3). The chamber is flushed with pure oxygen to regenerate the sensors in the next step. Due to the filling and flushing of the measurement chamber, peak-shaped measurement signals are obtained every 5 min. The analog measurement signal was converted to a digital signal by a microcontroller and forwarded to the computer interface, where it was integrated and processed with a prepared script with Matlab.

The gas sensor output is used as the source for the independent variables, while pH and TTA are the dependent variables whose relation to the independent variable should be predicted. We analyzed the raw signal from the gas sensor arrays (TGS 822, TGS 813, and MQ3) in a Matlab script designated to extract the peak height and area of the peak in each five-minute interval. From the feature extraction, six independent variables (i.e., 3 sensors * 2 features per sensor) were obtained. Figure 3 visualizes the feature extraction from the raw data for the whole fermentation of F5. Three different methods for the determination of the independent variables were used.

The independent variables were inserted in the same procedure to implement a process model. For this, the measurement values of the dependent and independent variables were used in a process regression that correlates the inputs to supply parameters for predicting the behavior of pH and acidity for their respective fermentation. The established parameters were tied to the corresponding variables in the regression formula. With Excel’s solver function, the parameters were minimized, expressing the correlation of the corresponding variables. Every five-minute interval, a reference point was determined that would be used for the error calculation. Due to fermentation runtimes of 10 h, an initial vector of 120 values was created for every fermentation. Each model estimation was evaluated by calculating the sum of squared errors (SSE), the root mean square error (RMSE), and the percentage error (% Error) of the RMSE that adjusted the error to the range of values. Additionally, regression curves correlating the predicted values to the measured values were created, and a coefficient of determination ($R^{2}$) was assigned. Three different methods for the determination of the independent variables were used and are explained separately in the following:The sensor features are filtered for the time corresponding to the taking of the offline sample (dependent variable). The offline values for pH and acidity and the corresponding GSA outputs were used as inputs for the regression equations. Two regression equations were established, one for each feature the regression was based on. For the sensor features, the feature values were adjusted by subtracting the baseline value of the GSA measurement from the feature value. The regression equation for the peak height and peak area regression are shown in Equations (Equation 2) and (Equation 3):
(2)$$C=K_{1}+K_{2}\left(PH_{TGS822}-BL\right)+K_{3}\left(PH_{TGS813}-BL\right)+K_{4}\left(PH_{MQ3}-BL\right)$$
(3)$$C=K_{5}+K_{6}\left(PA_{TGS822}-BL\right)+K_{7}\left(PA_{TGS813}-BL\right)+K_{8}\left(PA_{MQ3}-BL\right)$$
with *C* as the predicted value for the dependent variable, Kn as the regression parameters, *BL* as the baseline, *PH* for peak height, and *PA* for peak area. After evaluating the sensor features separately and combined in our calibrations, we decided to use peak height because it delivered better results. We refer to this method as the **sensor signal method**.In the second method, the independent variables were determined using a PCA script on the six extracted features. The features were transformed into two principal components, with the values along the main axis as the output. This reduced the dimensionality of the six features as a collection of variables while maintaining the same length of values in the data matrix. Analogous to the first approach, the regression model was created using the transformed values of the two principal components as independent variables. We refer to this method as the **PCA regression method**. The regression equation is shown in Equation (Equation 4):
(4)$$C=K_{9}+K_{10}PC1+K_{11}PC2$$For the third method, the raw data were split into datasets corresponding to the peak-shaped five-minute intervals extracted from the Matlab script. The feature extraction was not executed because enough data points per dataset had to remain for further data analysis. In the next step, a PCA script continuously analyzes all intervals of one fermentation to assign a score for each interval. The offline data were interpolated for each interval to correspond to the eigenvalues. The eigenvalues were analogous to the first approach as the independent variable for creating the process models. Only one principal component was considered an input for the regression model, because the first principal component had an explained variance over 99.5%. Therefore, the second principal component would add noise to the process model. Still, the initial vector for the model evaluation contains 120 values due to the transformation of the 5 min intervals. We refer to this method as the **interval method**. Equation (Equation 5) shows the corresponding regression formula.
(5)$$C=K_{1}+K_{2}PC1$$

For certain combinations of process models, we performed a validation by inserting the model parameters of one model into the regression equation of a validation set. The ability of the model parameters to predict the behavior of another process model is evaluated using SSE, RMSE, and the percentage model error.

### 2.3. Experiment Design

This section describes the procedure and the design of our experiments for analyzing the performance of the gas sensor array system. The appendix provides an overview of the used instruments and materials (cf. Appendix A).

The sourdough was prepared with three different flours: two rice flours—one of them a white flour (Heimatsmühle) and the other a wholegrain flour (Heimatsmühle)—and a white wheat flour (Rettenmaier Mühle). As a starter, the “Reinzucht-Sauerteig Reis” from Böcker was used. The moisture content was about 58%, and the pH was about 3.7. It was stored at 4 °C–6 °C.

A total of 16 fermentations, named F1 to F16, were carried out to provide a variance of the fermentation conditions. The experimental design measured each flour twice at 28 °C and 32 °C with a DY of 200. For the validation, each flour was additionally measured at 30 °C with a DY of 200. Table 1 shows the used flour and starter batch labels. Flour A was measured thrice at 28 °C, because it was unclear whether enough data points were available after the GSA system crashed. Table 2 shows a measurement scheme with the temperature and flour combinations.

Figure 4 shows the experimental set-up of the sourdough fermentation. To heat the stainless-steel fermenter, (1) a water bath (2) (Fisions) was connected to the water inlet and outlet connections of the fermenter and set to the desired temperature. The outlets on the side of the fermenter were sealed; the lowest outlet (3) was intended to take offline samples of the sourdough. The lid to close the fermenter on the top side was connected with an impeller mixer (4) (Hydro-Mec) and an engine (5). The mixer was set to the second speed level. Two of three outlets were covered on the lid and one connected to the gas sensor module (6) that leads the exhaust gas from the fermenter to the gas sensor.

At the beginning and every consecutive hour of each fermentation, the pH and the TTA were measured offline. The pH and TTA were measured with a pH meter (Xylem) and pH electrodes (VWR, Xylem). In the appendix, we describe the procedures for measuring pH and TTA in detail (cf. Appendix B). These reliable manual measurements a using standard procedure provide a reference for the calibration, using the values measured with the gas sensor array system. To illustrate the fermentation process, we included Figure A1 with the change in pH and TTA from F9 in the Appendix A.

## 3. Results

In what follows, we present the results of the analysis of the measurements with the gas sensor.

### 3.1. Process Model Evaluation

To analyze the process models’ accuracy, we measured the error evaluation for SSE, RMSE, and RMSE percentage error. The results are shown in the order of signal sensor, PCA regression, and interval methods.

**pH Sensor Signal Method.** Table 3 shows the results of the pH sensor signal method. The features of peak height and peak area were carried out as separate process models. After testing the combinations of the two features on several fermentations, this approach was disregarded due to higher errors than the single-feature models. Except for fermentations F14 and F15 of the pH sensor signal method, each percentage error stayed under 10%. Comparing the peak height with the peak area model showed that the peak height model had a lower error rate. Due to the adjustment of the percentage error to the range of values, a comparison of the pH and TTA model errors was possible. Although the TTA model does not have outliers like fermentations F14 and F15 from the pH model, its average error is higher than the pH models.

**PCA Regression Method.** The process models for the PCA regression method were carried out by grouping the 28 °C and 32 °C fermentations of one flour type in their temperature and combining them. The grouping of the fermentations was decided after detecting that the error of the grouped data resulted in lower error rates than the single-fermentation models. The grouped fermentations are titled in Table 4 as the abbreviations for the flour type (A, B, and C) and temperature, in which 28 °C and 32 °C contain the two respective fermentations. The combination is indicated as the abbreviation with an asterisk (*), and the two fermentations from each temperature are used in parentheses. The results of the error calculation for the PCA regression model show mostly errors under 10% with few exceptions, namely the temperature combination of A with a percentage error of 16% for TTA and the 32 °C grouped data from B with a percentage error of 10% for TTA. The last outlier was the temperature combination of C with a percentage error of 12% for pH.

**Interval Method.** The last method to be evaluated for the model’s error is the interval method. Table 5 presents the model errors for fermentations F1 to F16. The interval method had three major percentage model errors in F7 and F11, with errors over 15%. Model errors between 10% and 15% were detected in F2, F3, F7, F12, F14, and F16. Every model with a major pH or TTA error also had at least 10% to 15% in the other criteria. This was not the case if a model had an error between 10% and 15%. The grouped fermentations for the interval method in Table 6 are titled in the same way as the ones from the PCA regression method in Table 4.

The grouped temperature models showed a higher error than the single-fermentation models. Especially, the 32 °C model of flour type B and the 28 °C model of flour type C had percentage errors of over 15%.

### 3.2. Coefficient of Determination ($R^{2})$

The determination of the $R^{2}$ was only possible for the sensor signal models, because the integration of the PCA in the modulation methods caused a higher scattering of the predicted values. The determined $R^{2}$ values for the sensor signal models of fermentation F1 to F16 are shown in Table 7.

The $R^{2}$ values for the peak height models were generally higher than the ones for the peak area models. The comparison of pH and TTA models showed that none had consistently better values of $R^{2}$.

### 3.3. Validation of the Models

The validation was carried out by inserting the model parameters of one model in the regression equation of another model. For example, F4 with F8 means that the model parameters from F8 were inserted in the regression equation of F4. Table 8, Table 9 and Table 10 show the SSE, RMSE, and RMSE percentage model error of the validation models for the three different model methods.

**pH Sensor Signal Method.** The validations for the sensor signal method were only carried out for the peak height feature due to a lower error rate than the peak area feature. To evaluate if the models can describe the process within their flour type and temperature, the model parameters of these fermentations were inserted into each other. The validations showed a high percentage error, except for the model parameters of F4 and F10. In comparison, the percentage errors for TTA are lower than for pH.

**PCA Regression Method.** The validations for the PCA regression method were carried out by inserting the model parameters of the grouped temperature models into the 30°C calibration of their respective flour type.

The validations for the PCA regression method had a low error rate than the validations of the sensor signal method within their temperature. The percentage error is lower in the validations for the pH than it is for the TTA validations.

**Interval Method.** The validations for the interval method were based on examining different combinations between the grouped temperatures and the 30 °C validation temperature. The interval method showed a high scattering effect of the predicted values compared with the other methods. The model parameters had a high range of −200 to 200, which made a precise estimation of the validation sets difficult. Further validations were disregarded due to their high error rate.

## 4. Discussion

The results of the measurements using our evaluation settings indicate that for both pH and TTA, the best validation predictions were obtained by the PCA regression method. We interpret and discuss the results in detail in this section. Further, we explain the identified threats to validity.

### 4.1. Offline Data of Sourdough Fermentation

We performed offline measurements in the laboratory to confirm the results measured with the GSA system. The pH and TTA values behaved mostly as expected. For the pH values, each fermentation, except for F4, F12, and F14, had a sigmoid downward trend opposed to the growth of the microorganisms. The other ones showed a more parabolic behavior. This could have resulted from a faster or slower accommodation of the starter microorganisms in these fermentations. The TTA values showed an almost linear increase. The different flour types and temperatures influenced the fermentation as supposed. A higher temperature resulted in a faster process regarding pH and TTA. The starting pH of the white flours was lower than the wholegrain flour, while the starting TTA of the wholegrain flour was higher than the white flours. This could be explained by the degree of grinding and the state of the compounds in the flours. From the high degree of grinding, the white flours exhibit damaged starch molecules. These release more directly fermentable substrates that contribute to faster pH lowering. On the other hand, wholegrain flour still contained more protein and enzymes that exhibit a buffering effect [23]. Bolarinwa et al. [11] investigated the influence of temperature and fermentation time on pH and TTA by creating a prediction model using response surface methodology. In comparison, we varied temperature and flour types to increase the variance for the calibration input and then examined the prediction performance for our target variables. The $R^{2}$ for their model is 0.88 for pH and 0.887 for TTA, while our models range from 0.94 to 0.998 for pH and 0.947 to 0.994 for TTA. Other evaluation criteria were not stated and cannot be compared.

The HPLC results contribute insight into the metabolic processes during the fermentations. Each flour type had a different composition of sugars (glucose and maltose), but the microorganisms mainly converted glucose. This was visible by examining the change in concentrations of maltose between the wholegrain rice flour and the wheat flour. The wholegrain flour showed a low maltose concentration and was still not completely converted. Conversely, wheat flour showed a high concentration of maltose, but its concentration did not change drastically. In contrast, the concentration of glucose increased and decreased during the fermentation. The assumption that the content of organic acids was higher in wholegrain flour than in white flour was confirmed by the HPLC. Due to the lower grinding of the flour, more enzymes were available to convert a broader range of substrates. The results follow this trend.

### 4.2. Process Models

The process models for the sensor signal method had an adjusted percentage error of less than 10% for all models of the fermentations, except the ones for F14 and F15 of the peak area. They had a high coefficient of determination for predicting their fermentation but showed high errors for the validation with their paired temperature of the same flour type. The PCA regression method had low errors for the grouped temperature models and, compared with the sensor signal method, lower error values for the validations. The interval method contributed mixed error rates for the individual fermentation models and a percentage error of over 15%, except for one combination for the validations that had been carried out.

The aim for the validation errors was to be less than 10%; these values were mainly not reached. The best results for the validation models were achieved with the PCA regression method. Several possibilities could cause the validation of the methods not to satisfy the error requirements for the broad range of process models. The first reason is tied to the sensor signal and PCA regression method, because the interval method did not use feature extraction. The feature extraction of the peak area included a certain amount of noise. By adjusting the script further on the steps that capture the interval, less noise could be incorporated in the peak area feature. A second reason can be found in the GSA system. The already mentioned system crashes during the fermentations led to data gaps in the gas sensor online data. These gaps were carried over to later operations like feature extraction or PCA. There, the predicted values deviate noticeably after the data gap. This, in turn, impacts the error rate of the process model. The third reason is the high scattering of the PCA values omitted from the interval method. In this method, scores were assigned for every five-minute interval. Still, due to the similarity of the interval inputs from the raw data, the PCA contributed slight variance inside the range of the intervals. It must also be considered that the influence of the fermentation temperature changes the composition of the gas phase from the exhaust gas and, therefore, the signal response of the sensors. This could affect the prediction ability of models that combine data from different fermentation temperatures. To ensure the ability to monitor the process online, the PCA can be integrated into the data collection during measurement. In this work, the methods were evaluated on the measured data, but with a verified method, the data can be processed online without time delay.

Many studies use soft sensor models in fermentation monitoring. Mei et al. [19] proposed a multimodel method using Gaussian process regression and PCA to construct a soft sensor for fermentation processes to estimate biomass concentration. Similar to our approach, they used PCA to extract features and then integrated them into regression models. While our approach extracted the principal components to implement different methods into the parametric regression models, their approach calculates weights from submodel variance to combine into a final prediction model. We combined a data-driven approach with local models for the best prediction performance. But there are no approaches to building soft-sensing (pH/TTA) monitoring models for sourdough fermentation, so we cannot compare our results directly.

### 4.3. Threats to Validity

One limitation is the applicability of the results regarding other flour types and their influence on the final product. With supporting measurements, e.g., rheology for the characterization of the sourdough, it would become apparent if the models are applicable to sourdoughs from different flour types. Similarly, sensory evaluation has yet to be carried out to verify if products from the monitored sourdough would satisfy the requirements of consumers. These steps can be implemented to continue this work to ensure the relevance of the developed models. To justify replacing the methods used to measure pH and TTA with our models, we need to compare the estimated errors of traditional methods (i.e., using a pH electrode to measure the pH and TTA) with the errors that occur in our models. While the error of the pH measurement device is supposed to have an accuracy of ±0.1 pH units, it is still influenced by several factors like fermentation conditions, electrode calibration, preparation, and withdrawal of the samples. An estimated error of 5% for our model, which specific validations achieved, can be achieved by tuning the method in accuracy with more input data and a streamlined method. This would allow our method to be in the same margin as the traditional method while being less susceptible to errors occurring during the measurement. The model error at 5% would still be higher than the error of the traditional method. However, the increase in accuracy by tuning cannot be estimated safely, which can result in higher or lower degrees of accuracy improvement.

To specify, the results of the process models could have been improved by carrying out more fermentations with the same substrate and temperature to build a sufficient data foundation for the model creation and further data analysis operations. In this vein, using wheat flour instead of a third rice flour was too ambitious. An experimental plan with a third rice flour or more fermentations with the same substrates and temperatures might have led to more balanced and coherent datasets. With more extensive training and validation datasets, there are opportunities to use machine learning operations like neural networks that improve the prediction capability of the GSA. A reordering of the measurement data can improve the robustness of the models by making the features invariant to temperature and flour type. Using neural networks with GSA already proved successful in the contribution from Omatu and Yano [24]. By applying neural networks on time series data from GSA, they achieved a classification rate of 89% to 96% for tea and coffee odors.

One important factor to consider is the temperature drift caused by the temperature change in the environment. This greatly affects the precision and measurement stability of the gas sensors. As a solution, the approach by Xu et al. [25] can be used in future experiments. They proposed a compensation training method based on random forest, which improved the accuracy of the GSA by about 1%.

The use of the GSA system to predict process variables is promising, and with the research and implementation of the correct methods and tools, it can be a practical and easily implementable sensor. A possible idea for the future use of the GSA system would be to measure fermentations at the same process conditions for different time durations. With the help of a forecast algorithm, parameters can be adjusted, and an automated signal loop could be established. Tudu et al. [26] showed that a forecasting approach for the peak prediction of a black tea fermentation process is possible. They used a similar GSA system set-up to detect a peak representing the optimal fermentation time. It will be more challenging to align the total time series data to the optimization goal than to find a specific optimization peak. Still, with the variety of tools in the machine learning field, it is reasonable to accomplish. Using the GSA system could also deliver enough data for modeling the fermentation process as a digital twin [27].

## 5. Conclusions

For ensuring high-quality sourdough, monitoring the fermentation is essential. Relevant characteristics include pH and the degree of acidity measured as TTA. A time- and cost-intensive offline measurement of these variables can be avoided by utilizing online gas measurements in prediction models. In this paper, we describe a gas sensor array (GSA) system that can monitor the fermentation process of sourdough online. We used the obtained data to correlate the gas data with offline measurement values of the process variables. Three methods were tested to use the extracted features from GSA to create the prediction models.

The results indicate that the online measurement of exhaust gas from sourdough fermentation with gas sensor arrays can be a cheap and efficient application to predict pH and TTA. The work also showed that the data must be processed thoroughly and with a suitable method to achieve proper prediction performance.

In comparison with other approaches of fermentation monitoring, this approach needs just a simple and cheap set-up. The analysis of the data is conducted during the fermentation in real time. Measuring the variables (especially the pH values) directly might result in initially lower errors of measurements. Further, these approaches require less training data and, hence, less fermentations for generating training data have to be carried out. Moreover, the advantage of our approach is that the error can be decreased by training, and a noninvasive online fermentation monitoring model can be implemented.

The further steps to continue this work first include an analysis of a prototype that applies the PCA with real-time data. As the scope of this work was to identify suitable analysis approaches, this has not yet been carried out. Second, we plan to increase fermentation measurements to include machine learning operations in the model development reliably and add supporting measurements to characterize sourdough from different sources and at specific process parameters. Furthermore, a sensory evaluation must be added to guarantee the quality of bread products from the monitored sourdough.

## Figures and Tables

**Figure 1 sensors-23-07681-f001:**
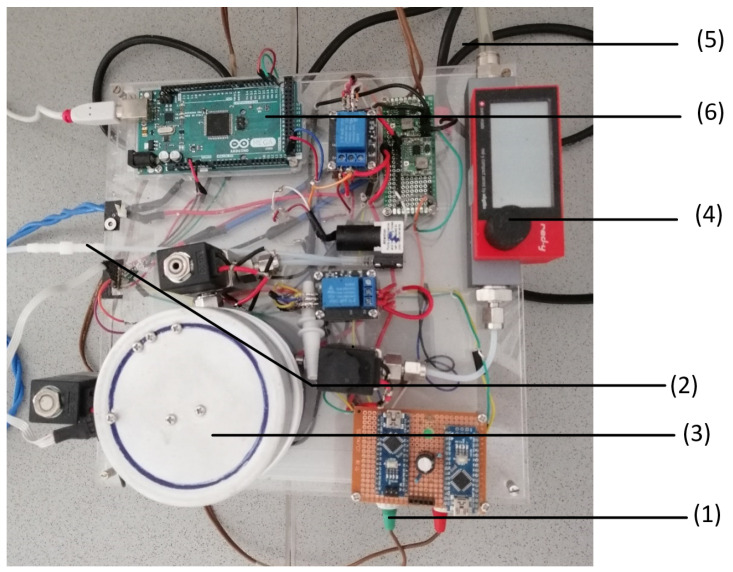
Set-up of the gas sensor array (GSA) system: The GSA contains a DC/AC converter (1), a tube connection from the oxygen gas cylinder (2), the gas measurement chamber (3), a flow meter (4), a tube connection from the bioreactor (5), and the Arduino microcontroller (6).

**Figure 2 sensors-23-07681-f002:**
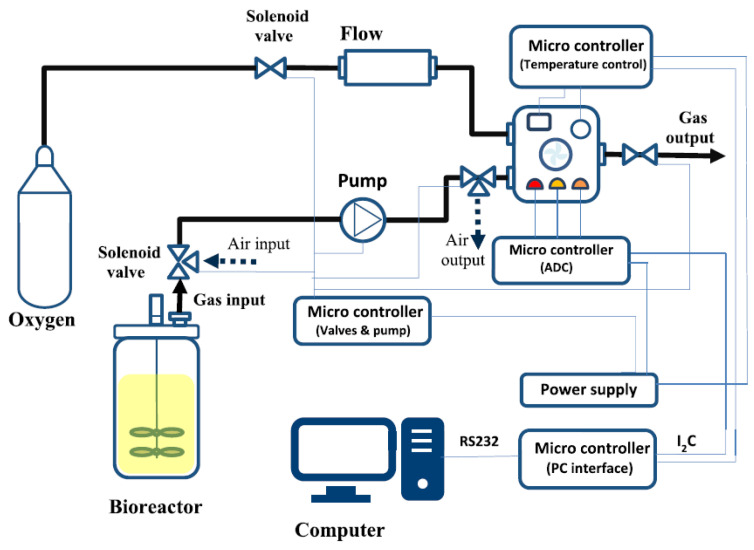
Schematic diagram of the GSA system [22] and its integration into the fermentation setup.

**Figure 3 sensors-23-07681-f003:**
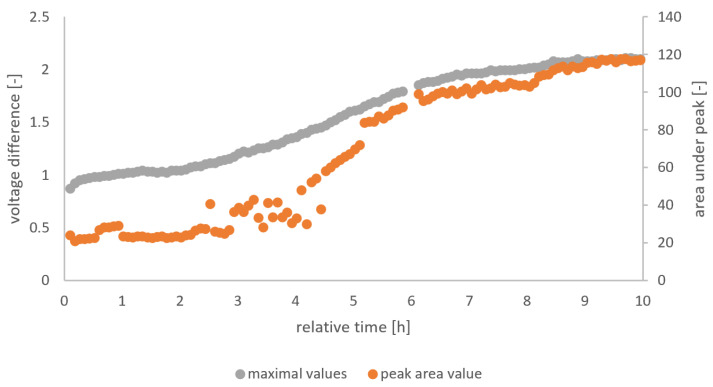
Peak height and area values from the feature extraction of MQ3 of F5.

**Figure 4 sensors-23-07681-f004:**
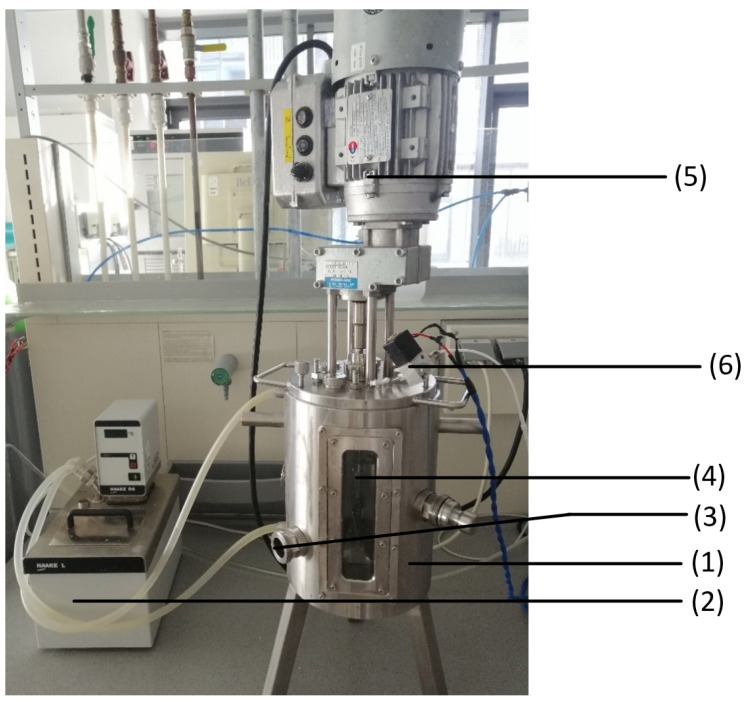
Experimental set-up of the sourdough fermentation: The set-up contains a stainless-steel fermenter (1), a water bath (2), an offline sample outlet (3), an impeller mixer (4), an engine (5), and the connection to the gas sensor module (6).

**Table 1 sensors-23-07681-t001:** Labels for the used flours and starter batches.

	Explanation
A	Wholegrain rice flour (Heimatsmühle)
B	White wheat flour (Rettenmeier Mühle)
C	White rice flour (Heimatsmühle)
s1	Starter batch 1 (3 November 2021–11 November 2021)
s2	Starter batch 2 (16 November 2021–23 November 2021)
s3	Starter batch 3 (24 November 2021–30 November 2021)
s4	Starter batch 4 (1 December 2021–6 December 2021)

**Table 2 sensors-23-07681-t002:** Experimental design for the fermentation measurements of temperature, flour type, and starter batch at a starter amount of 15% and DY of 200.

Fermentation	Temperature [°C]	Flour Type	Starter Batch
F1	28	A	s1
F2	32	A	s1
F3	30	A	s1
F4	28	A	s1
F5	32	A	s2
F6	28	A	s2
F7	32	B	s2
F8	28	B	s2
F9	32	B	s3
F10	28	B	s3
F11	30	B	s3
F12	32	C	s3
F13	28	C	s3
F14	32	C	s4
F15	28	C	s4
F16	30	C	s4

**Table 3 sensors-23-07681-t003:** Calculated SSE, RMSE, and percentage model errors of the pH sensor signal method for fermentations F1 to F16.

Fermentation	pH					
	Peak Height			Peak Area		
	SSE (-)	RMSE (-)	% Error	SSE (mL/10 g)	RMSE (mL/10 g)	% Error
F1	0.02	0.04	2.2	0.06	0.08	3.1
F2	0.04	0.07	3.3	0.06	0.08	3.8
F3	0.07	0.08	4.0	0.04	0.06	2.5
F4	0.11	0.09	4.7	0.24	0.14	7.1
F5	0.06	0.08	4.0	0.22	0.15	7.4
F6	0.02	0.04	1.9	0.03	0.06	2.8
F7	0.16	0.12	6.3	0.19	0.14	6.9
F8	0.05	0.07	3.7	0.03	0.06	2.9
F9	0.04	0.07	3.3	0.06	0.08	3.8
F10	0.03	0.05	2.5	0.01	0.03	1.4
F11	0.10	0.1	5.1	0.27	0.17	8.3
F12	0.04	0.07	2.6	0.20	0.14	7.1
F13	0.06	0.08	4.0	0.08	0.09	4.4
F14	0.11	0.1	5.2	0.60	0.25	12
F15	0.10	0.1	4.9	0.45	0.21	11
F16	0.03	0.05	2.6	0.52	0.23	9.2
	TTA					
F1	0.61	0.25	3.5	1.2	0.35	7.7
F2	5.5	0.74	7.4	3.8	0.62	6.2
F3	2.44	0.47	5.9	2.84	0.53	5.3
F4	1.77	0.38	6.4	2.16	0.42	7.1
F5	0.98	0.31	3.1	1.54	0.39	3.9
F6	0.48	0.21	2.8	1.16	0.34	4.5
F7	1.53	0.39	4.3	2.71	0.52	5.8
F8	0.69	0.26	3.8	3.09	0.56	7.9
F9	0.66	0.26	2.9	1.25	0.35	3.9
F10	1.11	0.33	5.1	1.68	0.41	6.3
F11	3.43	0.59	6.5	4.07	0.64	7.1
F12	0.67	0.26	5.7	1.54	0.39	6.0
F13	1.89	0.43	8.7	2.18	0.47	9.3
F14	0.3	0.17	2.7	1.54	0.39	6.0
F15	0.57	0.24	5.3	1.53	0.39	8.7
F16	0.32	0.18	3.6	1.15	0.34	7.5

**Table 4 sensors-23-07681-t004:** Calculated SSE, RMSE, and percentage model errors of the PCA regression method for the grouped 28 °C, 32 °C, and combined fermentations (assigned with the asterisk) from one flour type.

Fermentation	pH			TTA		
	SSE (-)	RMSE (-)	% Error	SSE (mL/10 g)	RMSE (mL/10 g)	% Error
A 28 °C	0.16	0.08	4.2	5.68	0.51	7.3
A 32 °C	0.17	0.09	4.4	4.43	0.45	6.4
A * (F2/F6)	0.5	0.15	7.5	26.13	1.09	16
B 28 °C	0.13	0.08	3.9	6.03	0.52	7.5
B 32 °C	0.29	0.12	5.8	10.84	0.7	10
B * (F7/F10)	0.38	0.13	6.6	6.32	0.54	7.7
C 28 °C	0.16	0.08	4.2	5.68	0.51	7.3
C 32 °C	0.17	0.09	4.4	4.43	0.45	6.4
C * (F13/F16)	1.21	0.23	12	1.33	0.25	3.5

**Table 5 sensors-23-07681-t005:** Calculated SSE, RMSE, and percentage model errors of the interval method for fermentations F1 to F16.

Fermentation	pH			TTA		
	SSE (-)	RMSE (-)	% Error	SSE (mL/10 g)	RMSE (mL/10 g)	% Error
F1	2.48	0.14	7.1	54.67	0.67	9.6
F2	3.46	0.17	8.4	62.27	0.71	10
F3	3.40	0.17	8.4	93.91	0.88	13
F4	2.14	0.12	6.1	60.77	0.65	9.2
F5	2.60	0.15	7.3	31.69	0.51	7.3
F6	3.98	0.18	9.0	48.02	0.63	9.0
F7	13.01	0.33	16	104.66	0.93	13
F8	2.55	0.14	7.2	17.65	0.38	5.4
F9	1.93	0.13	6.3	58.54	0.69	9.9
F10	1.62	0.12	5.8	6.74	0.24	3.4
F11	22.16	0.43	21	214.30	1.33	19
F12	5.18	0.21	10	53.77	0.66	9.5
F13	2.33	0.14	6.9	39.41	0.57	8.1
F14	5.00	0.21	11	16.42	0.38	5.5
F15	3.02	0.16	7.9	15.41	0.36	5.1
F16	5.51	0.21	11	13.45	0.33	4.7

**Table 6 sensors-23-07681-t006:** Calculated SSE, RMSE, and percentage model errors of the interval method for the grouped 28 °C and 32 °C fermentations from one flour type.

Fermentation	pH			TTA		
	SSE (-)	RMSE (-)	% Error	SSE (mL/10 g)	RMSE (mL/10 g)	% Error
A 28 °C	7.20	0.17	8.6	152.73	0.79	11
A 32 °C	10.24	0.21	10	120.76	0.70	10
B 28 °C	8.24	0.18	9.2	110.11	0.67	9.6
B 32 °C	25.68	0.32	16	502.25	1.43	21
C 28 °C	34.48	0.39	20	280,00	1.12	16
C 32 °C	15.96	0.26	13	170.31	0.85	12

**Table 7 sensors-23-07681-t007:** Results for $R^{2}$ from the pH and TTA sensor signal method for fermentations F1 to F16.

Fermentation	$R^{2}$ [-]			
	pH		TTA	
	Peak Height	Peak Area	Peak Height	Peak Area
F1	0.995	0.986	0.991	0.982
F2	0.993	0.990	0.952	0.969
F3	0.984	0.971	0.990	0.967
F4	0.978	0.952	0.972	0.965
F5	0.987	0.956	0.993	0.989
F6	0.997	0.993	0.994	0.985
F7	0.969	0.961	0.988	0.978
F8	0.987	0.992	0.988	0.947
F9	0.991	0.989	0.994	0.989
F10	0.993	0.998	0.98	0.969
F11	0.977	0.94	0.967	0.961
F12	0.988	0.944	0.988	0.972
F13	0.981	0.976	0.943	0.934
F14	0.969	0.827	0.994	0.971
F15	0.969	0.854	0.98	0.947
F16	0.993	0.848	0.991	0.969

**Table 8 sensors-23-07681-t008:** Results of SSE, RMSE, and percentage error for the validations of the sensor signal method from the fermentations of each flour type with the same temperature.

Validations	pH			TTA		
	SSE (-)	RMSE (-)	% Error	SSE (mL/10 g)	RMSE (mL/10 g)	% Error
A 32 °C						
F5 with F2	2.30	0.48	25	24.16	1.55	15
F2 with F5	2.80	0.53	26	65.61	2.56	26
A 28 °C						
F4 with F1	1.35	0.34	17	18.14	1.23	20
F1 with F4	0.17	0.13	6.5	3.43	0.59	8.4
B 32 °C						
F9 with F7	0.88	0.30	14	22.91	1.51	17
F7 with F9	1.75	0.42	21	14.59	1.21	13
B 28 °C						
F10 with F8	0.59	0.24	12	13.53	1.16	18
F8 with F10	0.40	0.20	9.9	4.13	0.64	9.2
C 32 °C						
F14 with F12	1.09	0.33	16	13.18	1.15	14
F12 with F14	4.67	0.68	27	4.12	0.64	14
C 28 °C						
F13	1.17	0.34	17	5.86	0.77	17
F15	3.46	0.59	29	8.65	0.93	19

**Table 9 sensors-23-07681-t009:** SSE, RMSE, and percentage error results for validating the PCA regression method from the grouped temperature models in the 30 °C validation set (combined fermentations assigned with the asterisk from one flour type).

Validations	pH			TTA		
	SSE (-)	RMSE (-)	% Error	SSE (mL/10 g)	RMSE (mL/10 g)	% Error
Validation of A 30 °C with						
A 28 °C	0.19	0.13	6.5	7.13	0.8	10
A 32 °C	0.75	0.26	13	25.35	1.52	19
A * (F2/F6)	0.5	0.15	7.5	8.75	0.89	11
Validation of B 30 °C with						
B 28 °C	1.17	0.33	16	28.71	1.62	20
B 32 °C	0.38	0.19	9.3	1.84	0.41	5.1
B * (F7/F10)	0.76	0.26	13	29.27	1.63	20
Validation of C 30 °C with						
C 28 °C	0.19	0.13	6.5	7.13	0.8	10
C 32 °C	0.68	0.25	12	9.98	0.95	12
C * (F13/F16)	0.8	0.27	13	2.73	0.5	6.2

**Table 10 sensors-23-07681-t010:** Results of the SSE, RMSE, and percentage error for the validation of the interval method.

Validations	pH			TTA		
	SSE (-)	RMSE (-)	% Error	SSE (mL/10 g)	RMSE (mL/10 g)	% Error
A 30 °C with A 32 °C	31.01	0.50	25	927.55	2.76	39
C 32 °C with A 32 °C	31.40	0.37	18	7869.90	5.79	83
B 30 °C with B 28 °C	29.66	0.49	25	433.64	1.89	27
B 28 °C with C 28 °C	52.66	0.46	23	674.81	1.66	24
C 30 °C with C 28 °C	13.01	0.33	16	86.39	0.84	12

## Data Availability

Data is available upon personal request at the corresponding author.

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
