# Peer review of "Online Monitoring of Sourdough Fermentation Using a Gas Sensor Array with Multivariate Data Analysis"

_sensors, 2023, doi:10.3390/s23187681_

Round 1
Reviewer 1 Report
The manuscript lacks the correct structure. Please completely reformat the manuscript according to the journal guidelines
Example: The manuscript lacks the correct structure :
Introduction, Materials and Methods, Results, Discussion
Please READ published papers and learn how to format the manuscript correctly
This work establishes a gas sensor array system to predict acidity and total titratable acidity in a Sourdough fermentation system. The work adopts multiple methods to predict these two parameters but does not discuss the results relative to previous studies. The manuscript must be reformatted to include only: Introduction, Methods and Material, Results, Discussion, and Conclusion. Any other section should be a subsection. That said, the Introduction and Background must be merged and shortened to 5-6 paragraphs, excluding the objective paragraph, so a maximum of 7 paragraphs. This is not a report. This is a manuscript. I urge the authors to read other published papers to learn from their formatting. In addition, please check the language, and ensure everything is written in the correct tense.
Specific comments:
1) Lines 1-12: Please add some stats from your results.
2) Lines 37 through 40: Please do not use number points here, please rewrite within the paragraph. Also, do not add what the work does here. Add this to the last paragraph of the Introduction. The objective of this work. The Intro and Background must be combined and restructured.
3) Line 36: Why is does PH need to be predicted? Why not use a PH sensor?
4) Lines 41 through 47: This must be moved to results. The introduction should not have any results.
5) Lines 71-74: If equation 1 is used in this work for calculation, please move to the Methods section and provide a citation.
6) Lines 54-79: the text should be reduced. Why is there a need to provide so much detail? Be concise and refer the reader to citations for additional reading.
7) Line 79 (Gas sensors) starts with no introduction. I just read a whole paragraph about Sourdough, fermentation and level of acidity. The authors then switched to VOC sensors, but the authors did not mention anything about VOC generation! All this background but nothing about VOC! Need to establish the need for VOC sensors and make this a new paragraph in the introduction.
8) Lines 94-118: This paragraph should be mentioned before mentioning the VOC sensors and should be reduced in text and part of the Introduction section. Please make this concise, do not write long paragraphs for the Introduction or in general. Break down ideas! Remember, 7 paragraphs max!
9) Lines 41-53 and Lines 119-125: Combine these paragraphs into one paragraph, the objective paragraph and no need to explain how your manuscript is structured. This should be your structure: Introduction, Methods and Material, Results, Discussion, and Conclusion. So please remove lines 48-42.
10) Line 126: This is where Methods and Materials start, please make 4 a subsection.
11) Line 126-146: I am confused between Figure 1 and Figure 2. As described, one is a picture and the other is a schematic. However, Figure 2, looks like the schematic of the fermentation system and NOT the GSA system. Please clarify.
12) Line 150: “From the feature extraction, six independent variables (i.e., 3 sensors * 2 features per sensor) are obtained.” Should be in the past “sensor) were obtained”. Please check the Methods section for consistency.
13) Line 154: What reference sensors did you use to measure pH and acidity?
14) Lines 147- 162: Please explain who is the dependent and the independent variable before equations 2 and 3. Can you please make it clear .
15) Lines 226: it is important to mention in the manuscript what sensors were used to measure PH and acidity.
16) Lines 223-237: Where did you develop these values? Do you have a citation?
17) Line 229: This should be section 3. Results
18) Line 305: This should be section 4. Discussion
19) Lines 305-394: Please add a comparison of your work to others from the literature. How does your work compare to previously published studies in each section?
20) Line 305: This should be section 5. Conclusion
Author Response
First of all, we would like to thank the reviewer for taking time to provide a detailed revision. The comments and suggestions for improvement were of great value. We carefully considered all comments and implemented them into the paper to the best of our abilities. Passages that were possibly misunderstood were improved in clarity, enhancing legibility of the paper. We marked those changes in blue (except of minor adjustments due to spelling / grammar issues).
#R1-1 : The manuscript lacks the correct structure. Please completely reformat the manuscript according to the journal guidelines [..]
In addition, please check the language, and ensure everything is written in the correct tense.
We thank the reviewer for the information about the recommended structure. We were not aware of this strict requirement for the structure as we had already other papers in MDPI journals (including Sensors) with a similar structure and got never complaints. Further, when comparing to other papers in the journal, there are many other works that follow another structure.
Still, we addressed this comments and revised the structure (see also #R1-3, #R1-5, #R1-6, #R1-7, #R1-9, #R1-10, #R1-11, #R1-18, #R1-19, and #R1-21). However, we disagree that the “Objective” and “Stucture” paragraphs do not fit as we think it is a necessary part for a research paper.
We further checked the language using a corresponding language checking tool.
#R1-2: Lines 1-12: Please add some stats from your results.
We added the range of RMSE and R2 of the calibrations.
#R1-3: Lines 37 through 40: Please do not use number points here, please rewrite within the paragraph. Also, do not add what the work does here. Add this to the last paragraph of the Introduction. The objective of this work. The Intro and Background must be combined and restructured.
We removed the enumeration and changed the structure accordingly.
#R1-4: Line 36: Why is does PH need to be predicted? Why not use a PH sensor?
pH is difficult to measure in media with high solid contents, therefore an indirect prediction approach has advantages in a cost and assembly perspective.
#R1-5: Lines 41 through 47: This must be moved to results. The introduction should not have any results.
We changed the structure accordingly. Please see also comment #R1-1.
#R1-6: Lines 71-74: If equation 1 is used in this work for calculation, please move to the Methods section and provide a citation.
We changed the structure accordingly and moved the text to the “Methods and Materials” section. Please see also comment #R1-7. The citation has been added in line 109.
#R1-7: Lines 54-79: the text should be reduced. Why is there a need to provide so much detail? Be concise and refer the reader to citations for additional reading.
We reduced the text and shifted the paragraph to the “Methods and Materials” section. Please see also comment #R1-6.
However, as the journal that we target is not a food-specific journal, we think this is necessary background information that many readers of the journal usually do not have.
#R1-8: Line 79 (Gas sensors) starts with no introduction. I just read a whole paragraph about Sourdough, fermentation and level of acidity. The authors then switched to VOC sensors, but the authors did not mention anything about VOC generation! All this background but nothing about VOC! Need to establish the need for VOC sensors and make this a new paragraph in the introduction.
We revised the structure and text accordingly: Related work was shifted to the introduction, and the necessary background information to the “Methods and Materials” section.
#R1-9: Lines 94-118: This paragraph should be mentioned before mentioning the VOC sensors and should be reduced in text and part of the Introduction section. Please make this concise, do not write long paragraphs for the Introduction or in general. Break down ideas! Remember, 7 paragraphs max!
We integrated the paragraphs about related work into the paper’s introduction. Still, we think that it should be clearly described what the state of the art is, please see also the comments of the second reviewer (cf. #R2-1). So we need to balance these two contradicting views.
#R1-10: Lines 41-53 and Lines 119-125: Combine these paragraphs into one paragraph, the objective paragraph and no need to explain how your manuscript is structured. This should be your structure: Introduction, Methods and Material, Results, Discussion, and Conclusion. So please remove lines 48-42.
We revised the text accordingly. However, we kept the paragraph about the structure of the document as we think this is an elementary part of the introduction of a research paper. We did not understand to which lines the reviewer is referring to when mentioning lines 48-42, but we assume it’s the paragraph about the structure, which we will keep as mentioned.
#R1-11: Line 126: This is where Methods and Materials start, please make 4 a subsection.
We changed the structure accordingly. Please see also comment #R1-1.
#R1-12: Line 126-146: I am confused between Figure 1 and Figure 2. As described, one is a picture and the other is a schematic. However, Figure 2, looks like the schematic of the fermentation system and NOT the GSA system. Please clarify.
Indeed, Figure 2 shows how the GSA is integrated into the fermentation setup. Still, the figure focuses on the components of the GSA. We revised the text and the caption of the figure.
#R1-13: Line 150: “From the feature extraction, six independent variables (i.e., 3 sensors * 2 features per sensor) are obtained.” Should be in the past “sensor) were obtained”. Please check the Methods section for consistency.
We revised the text and checked the whole section regarding the used tense.
#R1-14: Line 154: What reference sensors did you use to measure pH and acidity?
Please see comment #R1-16, which also addresses this issue.
#R1-15: Lines 147- 162: Please explain who is the dependent and the independent variable before equations 2 and 3. Can you please make it clear .
The assignment of the independent and dependent variable has been clarified in lines 140-142.
#R1-16: Lines 226: it is important to mention in the manuscript what sensors were used to measure PH and acidity.
We added the used sensors for the pH and Acidity measurements and included a reference to the procedures of the measurement.
#R1-17: Lines 223-237: Where did you develop these values? Do you have a citation?
We removed the classification of error percentages, because the variance within studies does not allow for a general categorization.
#R1-18: Line 229: This should be section 3. Results
We changed the structure accordingly. Please see also comment #R1-1.
#R1-19: Line 305: This should be section 4. Discussion
We changed the structure accordingly. Please see also comment #R1-1.
#R1-20: Lines 305-394: Please add a comparison of your work to others from the literature. How does your work compare to previously published studies in each section?
We added two works from the literature to set our approach into relation to recent studies.
#R1-21: Line 305: This should be section 5. Conclusion
This comment contradicts the previous one (#R1-19) as the line number does not fit. Hence, we assumed you mean that the conclusion should be Section 5, which is the case now.
Reviewer 2 Report
The presented article is devoted to an actual technological problem - monitoring the fermentation of sourdough from various flour types, however, the manuscript needs serious revision for publication in the Sensors. The main comments on the material are as follows:
1. The literature review on the chosen problem is scarce, there are much more works on the use of sensor systems and arrays of sensors to assess fermentation and other processes, including the variants of data processing. For example, https://doi.org/10.3390/s20061771, https://doi.org/10.1016/j.aca.2013.09.048
2. From the presented material it is difficult to assess the scientific novelty and significance of the proposed solution. Principal component analysis is a traditional approach for dimension reduction and feature extraction from multivariate data. However, the meaning of the application is not entirely clear, initially the authors have three independent variables, which can be easily analyzed without multivariate data processing methods. Applying regression after principal component analysis is called principal component regression and is also a traditionally used approach. It is necessary to explain how the approach proposed by the authors differs from the existing one. The most interesting method is interval method, however, it is not clear why a five-minute interval was chosen, how many measurements were made during this time, etc.
3. When compiling the regression equations with peaks areas, as well as for the peaks heights, the baseline was subtracted. Was it the same value? Since there are no other explanations in the text, it seems that this is the same value, although this is mathematically incorrect, in this case it is better to calculate the peak area immediately in relation to the baseline, which can be taken into account when programming the calculations, in this case the resulting regression equations will be more accurate. In the article, it is necessary to provide at least one figure with the output curves of sensors during measurement.
4. The results obtained should be discussed from the standpoint of already known works in the field of data processing of metal oxide sensors. From this point of view, some results are quite obvious. Temperature is a very powerful external factor that significantly affects the sensor signal, and in order to improve the predictive properties of models, it would be interesting to take into account techniques for compensating the temperature drift of sensor signals (such as https://link.springer.com/chapter/10.1007/978-981-10-6373-2_14 ) when building models. Otherwise, the prediction error in most cases will a priori be higher when change the temperature.
5. Regressive equations are characterized by interpolation properties, the idea of ​​extrapolar properties of models needs to be clarified.
6. To assess the possibility of using the obtained regression equations, it is necessary to compare the prediction error by the model with the characteristics of the uncertainty of the predicted value, i.e. pH and TTA.
7. For the convenience of readers and evaluation of the results, explain how many measurements were made and what is the dimension of the initial matrix for calculating the SSE, RMSE of each model, since these characteristics directly depend on the number of repetitions.
Author Response
First of all, we would like to thank the reviewer for taking time to provide a detailed revision. The comments and suggestions for improvement were of great value. We carefully considered all comments and implemented them into the paper to the best of our abilities. Passages that were possibly misunderstood were improved in clarity, enhancing legibility of the paper. We marked those changes in blue (except of minor adjustments due to spelling / grammar issues).
#R2-1: The literature review on the chosen problem is scarce, there are much more works on the use of sensor systems and arrays of sensors to assess fermentation and other processes, including the variants of data processing. For example, https://doi.org/10.3390/s20061771, https://doi.org/10.1016/j.aca.2013.09.048
We extended the literature review with the mentioned reviews and added further work about food fermentation and soft sensor applications for fermentations that are mentioned by the reviews.
#R2-2: From the presented material it is difficult to assess the scientific novelty and significance of the proposed solution. Principal component analysis is a traditional approach for dimension reduction and feature extraction from multivariate data. However, the meaning of the application is not entirely clear, initially the authors have three independent variables, which can be easily analyzed without multivariate data processing methods. Applying regression after principal component analysis is called principal component regression and is also a traditionally used approach. It is necessary to explain how the approach proposed by the authors differs from the existing one. The most interesting method is interval method, however, it is not clear why a five-minute interval was chosen, how many measurements were made during this time, etc.
Indeed, our contribution cannot be seen on the methodological level, as we applied existing methods (such as PCA). However, the application of such data analytics in the domain of sourdough fermentation is new. We did small adjustment in the text to better reflect this.
#R2-3: When compiling the regression equations with peaks areas, as well as for the peaks heights, the baseline was subtracted. Was it the same value? Since there are no other explanations in the text, it seems that this is the same value, although this is mathematically incorrect, in this case it is better to calculate the peak area immediately in relation to the baseline, which can be taken into account when programming the calculations, in this case the resulting regression equations will be more accurate. In the article, it is necessary to provide at least one figure with the output curves of sensors during measurement.
Peak height and peak area were two different values, we adjusted both for the baseline and after evaluating them separately and combined in our first calibrations, we decided to take just the peak height feature because it delivered the best results (lines 172-174). In Figure 4, the sensor features from the measurement are included.
#R2-4: The results obtained should be discussed from the standpoint of already known works in the field of data processing of metal oxide sensors. From this point of view, some results are quite obvious. Temperature is a very powerful external factor that significantly affects the sensor signal, and in order to improve the predictive properties of models, it would be interesting to take into account techniques for compensating the temperature drift of sensor signals (such as https://link.springer.com/chapter/10.1007/978-981-10-6373-2_14) when building models. Otherwise, the prediction error in most cases will a priori be higher when change the temperature.
We added two works from the literature to set our approach into relation to recent studies. The problem of temperature drift has been acknowledged and was added to the threats to validity section (lines 400-404).
#R2-5: Regressive equations are characterized by interpolation properties, the idea of ​​extrapolar properties of models needs to be clarified.
Indeed, this was an error in the manuscript, as we target interpolation. The models might be also used for extrapolation of the values, which is out of scope of this work.
#R2-6: To assess the possibility of using the obtained regression equations, it is necessary to compare the prediction error by the model with the characteristics of the uncertainty of the predicted value, i.e. pH and TTA.
Unfortunately, we do not fully understand the reviewer’s concern. Do you mean with unertauinty a comparison to the offline measurements? This is hardly feasible, as those can only return point measurements and would be compared against continuos measurements.
#R2-7: For the convenience of readers and evaluation of the results, explain how many measurements were made and what is the dimension of the initial matrix for calculating the SSE, RMSE of each model, since these characteristics directly depend on the number of repetitions.
We added the amount of measurement points that are included for the models and error estimation in lines 156 and 194.
Round 2
Reviewer 1 Report
The authors have addressed my comments and I accept the manuscript.
Author Response
Thank you again for your valuable feedback.
Reviewer 2 Report
The corrections and addition, made by authors, have improved the understanding of the points of the research, however, there are still several fundamental issues with the manuscript that do not allow recommending it for publication in this form. Since the novelty of the article is the use of an array of commercially available metal oxide sensors to monitoring of the fermentation process, additional explanations are needed to reveal this novelty more fully.
There are comments and recommendation:
1. It follows from the title that fermentation monitoring is supposed to be controlled online, however, the most effective data processing method is the PCA regression method (line 295). However, this method requires the preliminary collection of a data matrix from which the values of the principal components can be obtained. Can the authors explain or suggest how long, in this case, the analysis will take along with data processing, and will this be enough to control the fermentation process in a timely manner?
2. From the description of model validation (lines 280-304), it follows that the cross-validation approach was applied, however, before applying the verification of regression equations, it is necessary to assess the adequacy of the model, for example, by the Fisher criterion, and to evaluate the significance of the regression coefficients, for example, by the Student's criterion or other criterion, which assess how the studied quantities have a relationship with each other. In addition, by the value of the regression coefficients, it can be seen which sensors make what contribution to the equation and whether all three sensors are needed or some can be excluded. If for some reason it is impossible to give the values of the obtained coefficients, then in the Discussion section it is desirable to discuss the results of checking the coefficients for significance and the data obtained. This point is directly related to the next comment.
3. The method validation section is redundant (lines 280-304), since when substituting the values of one process into another model, RMSE will increase by some value. Since SSE, RMSE and error correlate with each other, in my opinion, it would be more significant not to give RMSE values and errors for all models, but to indicate on the diagram the degree of increase in error for different types of flour and temperatures, for example, for A flour at temperature At 32C, the error increases on average by 11.5% (from 3.7 to 25.5%), and at 28C, by 8.5%, etc. And, in general, it is desirable to use more illustrative materials to describe the results obtained.
4. In section 4.3. When replacing methods or techniques in production, one of the important characteristics is the error in determining the required values. In this regard, when discussing the possibility and effectiveness of replacing methods for measuring pH and TTA using an array of sensors, it is necessary to consider the error in determining pH and TTA using traditional methods. For a positive assessment of the possibility of replacing one method with another, the errors must be comparable in magnitude and allow to determine with sufficient accuracy deviations from the normotypical process in production, or indicate options for achieving this accuracy, which is already partially described in this section.
5. In conclusion, it would be useful to indicate the advantages and disadvantages of the proposed approach when compared with other methods of fermentation control (analysis time, error, number of fermentation processes ensured by the uninterrupted operation of the device, etc.).
check the spelling and articles
Author Response
First of all, we would like to thank the reviewer for taking time to provide a detailed revision. The comments and suggestions for improvement were of great value. We carefully considered all comments and implemented them into the paper to the best of our abilities. Passages that were possibly misunderstood were improved in clarity, enhancing legibility of the paper. We marked those changes in green (except of minor adjustments due to spelling / grammar issues) and also kept the marking of the changes for the previous revision in blue.
1. It follows from the title that fermentation monitoring is supposed to be controlled online, however, the most effective data processing method is the PCA regression method (line 295). However, this method requires the preliminary collection of a data matrix from which the values of the principal components can be obtained. Can the authors explain or suggest how long, in this case, the analysis will take along with data processing, and will this be enough to control the fermentation process in a timely manner?
In this work, the methods have been evaluated on the measured data to investigate if the PCA can help to analyze the data. The next step would be to transfer the results to an online system. We assume the data can be processed online without timely delays as the PCA might require computational time to identify the relevant factors - later, we can observe those factors (see lines 369-372 and 452-454).
2. From the description of model validation (lines 280-304), it follows that the cross-validation approach was applied, however, before applying the verification of regression equations, it is necessary to assess the adequacy of the model, for example, by the Fisher criterion, and to evaluate the significance of the regression coefficients, for example, by the Student's criterion or other criterion, which assess how the studied quantities have a relationship with each other. In addition, by the value of the regression coefficients, it can be seen which sensors make what contribution to the equation and whether all three sensors are needed or some can be excluded. If for some reason it is impossible to give the values of the obtained coefficients, then in the Discussion section it is desirable to discuss the results of checking the coefficients for significance and the data obtained. This point is directly related to the next comment.
The idea for utilizing several gas sensors is to increase the variance and identify hidden relationships. A preselection would decrease the range of input variables and diminish the ability to include reverse characteristics. We tried different combinations to ensure that the included features showed reasonable results. The relation of studied quantities (pH, TTA, gas development during fermentation) is already established, and the regression coefficients include the sensor signals' information. Assessing the regression coefficient significance would not contribute to further information gained on the model.
3. The method validation section is redundant (lines 280-304), since when substituting the values of one process into another model, RMSE will increase by some value. Since SSE, RMSE and error correlate with each other, in my opinion, it would be more significant not to give RMSE values and errors for all models, but to indicate on the diagram the degree of increase in error for different types of flour and temperatures, for example, for A flour at temperature At 32C, the error increases on average by 11.5% (from 3.7 to 25.5%), and at 28C, by 8.5%, etc. And, in general, it is desirable to use more illustrative materials to describe the results obtained.
This paper does not aim to describe and analyze the influence of the experimental variables (Flour Type and Temperature). They are meant to find models to handle the inputs and predict the process. Hence, we did not include those aspects.
4. In section 4.3. When replacing methods or techniques in production, one of the important characteristics is the error in determining the required values. In this regard, when discussing the possibility and effectiveness of replacing methods for measuring pH and TTA using an array of sensors, it is necessary to consider the error in determining pH and TTA using traditional methods. For a positive assessment of the possibility of replacing one method with another, the errors must be comparable in magnitude and allow to determine with sufficient accuracy deviations from the normotypical process in production, or indicate options for achieving this accuracy, which is already partially described in this section.
We tried to improve the mentioned section concerning the reviewer’s comments (see lines 390-401).
5. In conclusion, it would be useful to indicate the advantages and disadvantages of the proposed approach when compared with other methods of fermentation control (analysis time, error, number of fermentation processes ensured by the uninterrupted operation of the device, etc.).
We added such a comparison to the conclusion section (see lines 445-450).